# Small Noncoding Regulatory RNAs from *Pseudomonas aeruginosa* and *Burkholderia cepacia* Complex

**DOI:** 10.3390/ijms19123759

**Published:** 2018-11-27

**Authors:** Tiago Pita, Joana R. Feliciano, Jorge H. Leitão

**Affiliations:** iBB-Institute for Bioengineering and Biosciences, Departmento de Bioengenharia, Instituto Superior Técnico, Universidade de Lisboa, Av. Rovisco Pais, Torre Sul, Piso 6, 1049-001 Lisbon, Portugal; tiagopita@tecnico.ulisboa.pt (T.P.); joana.feliciano@tecnico.ulisboa.pt (J.R.F.)

**Keywords:** cystic fibrosis, *Pseudomonas aeruginosa*, *Burkholderia cepacia* complex, small noncoding regulatory RNAs, pathogenicity

## Abstract

Cystic fibrosis (CF) is the most life-limiting autosomal recessive disorder in Caucasians. CF is characterized by abnormal viscous secretions that impair the function of several tissues, with chronic bacterial airway infections representing the major cause of early decease of these patients. *Pseudomonas aeruginosa* and bacteria from the *Burkholderia cepacia* complex (Bcc) are the leading pathogens of CF patients’ airways. A wide array of virulence factors is responsible for the success of infections caused by these bacteria, which have tightly regulated responses to the host environment. Small noncoding RNAs (sRNAs) are major regulatory molecules in these bacteria. Several approaches have been developed to study *P. aeruginosa* sRNAs, many of which were characterized as being involved in the virulence. On the other hand, the knowledge on Bcc sRNAs remains far behind. The purpose of this review is to update the knowledge on characterized sRNAs involved in *P. aeruginosa* virulence, as well as to compile data so far achieved on sRNAs from the Bcc and their possible roles on bacteria virulence.

## 1. Introduction: What are Bacterial sRNAs?

Regulation of gene expression in bacteria is achieved by a diversified system of multiple regulators, from proteins to RNA molecules. Riboswitches, T Boxes, and small noncoding regulatory RNAs (sRNAs) are among the RNA molecules or sequences that regulate the expression of genes. sRNAs were first identified in *Escherichia coli* in the 1960s as part of a large group of transcriptional regulators [1,2]. Since then, sRNAs have been shown as key regulators in bacteria [3], being capable of driving the fastest responses in the bacterial cell [4]. Ranging from 50 to 400 nucleotides long in size, sRNAs have been found as encoded widespread in the bacterial genomes, mainly in intergenic regions. For example, in enterobacterial genomes, sRNAs are estimated to represent approximately 200 to 300 genes, corresponding to ~5% of the number of protein-encoding genes [4,5].

sRNAs can be multipurpose regulators, targeting a wide range of molecular structures such as DNA/chromatin, proteins, other RNA molecules, and metabolites [6]. These regulatory sRNAs are therefore involved in the regulation of diverse cellular processes, including DNA assembly, plasmid replication, phage development, transcription, translation, peptidoglycan synthesis, and bacterial virulence [1,6]. In addition, sRNAs are also known for playing a role in cellular metabolism, like carbon and amino acid metabolism, iron homeostasis, quorum sensing (QS), and biofilm formation; stress responses including acid, osmotic, and oxidative stress responses; adaptation to growth conditions like high temperature and changes in molecular oxygen availability; as well as in mechanisms related to bacterial pathogenesis [7].

Although the vast majority of the so far characterized sRNAs do not encode proteins, a few can be partially translated originating small peptides, collectively known as dual-function small regulatory RNAs. These sRNAs can exert their regulation by acting as an RNA molecule or as a peptide. The peptide can play a similar regulatory role or a distinct one, capable of acting in a distinct metabolic pathway [8].

The best described sRNAs act by antisense base pairing with a specific target mRNA. These sRNAs can be cis- or trans-encoded, depending on their genome location in relation to the mRNA target [6]. Cis-encoded antisense RNAs share a full complementarity with their target mRNAs, often leading to the formation of a near perfect sRNA-mRNA duplex. Distinctly, trans-encoded antisense sRNAs only establish a partial complementarity with their target mRNAs, forming a partial duplex. Due to their lower specificity, trans-encoded sRNAs usually have multiple targets, being part of a more complex regulatory network in the cell. In both cases, the interaction between the antisense sRNA and target mRNA can affect translation negatively or positively [9].

The base pairing of the sRNAs can occur at different regions of the target mRNA, such as the 3′ untranslated region (3′-UTR), the coding region, or, like most of the cases, at the 5′ untranslated region (5′-UTR) where the ribosome accommodation usually occurs. The result of the interactions between the sRNA and the target mRNA can also differ, often leading to the suppression of gene translation by inhibition of ribosome interactions and/or by the induction of degradation of the target mRNA. However, activation of gene translation has also been reported. In this case, the interaction of the sRNA with the translation initiation region of the mRNA leads to a restructure of the mRNA conformation, exposing the previously occluded ribosome biding site (RBS) [6,10]. sRNAs can also act as stabilizers of target RNAs, avoiding their degradation. In addition, sRNAs can actually lead to the cleavage of mRNAs, resulting in one or two stable coding mRNAs [9].

Sometimes a sRNA can also become a target, when another RNA molecule (usually a mRNA) binds to the sRNA, trapping it. Known cases involve a sRNA that is constitutively expressed, and it is the expression of the trap-mRNA that will impair the regulatory function of the sRNA on its target [11,12]. Nevertheless, it has also been observed that some sRNAs bind to other sRNA, like the SroC of *Salmonella* that can sequester a given sRNA, impairing its activity [13].

The interaction between sRNAs and proteins usually leads to protein sequestration [9]. One of the best characterized target protein is CrsA, which has the ability to directly bind to the 5′ regions of mRNAs, repressing their translation [14]. Some sRNAs act like a sponge, sequestering this protein. This interaction activates the regulation of multiple pathways, related to carbon starvation and glycogen biosynthesis, affecting the physiology and virulence of several pathogenic bacteria [14]. Moreover, sRNAs can also bind to proteins in a more complex way, producing more complex outcomes, such as the modulation of enzymatic activity by inhibiting, activating or modifying the protein activity. The best known example is *E. coli* 6S sRNA, which binds to the housekeeping form of RNA polymerase (σ^70^-RNAP), modifying its activity [3]. Nevertheless, the best characterized proteins that interact with sRNAs are the Hfq-like RNA chaperones, which play a major role in the regulatory mechanism of sRNAs base pair interactions, by stabilizing the RNA molecules and mediating the interaction between sRNAs and their targets.

Proteins of the Hfq family are conserved among bacterial genomes, being present in approximately one half of the sequenced organisms, a fact that certainly reflects its importance for the bacterial cell [15,16]. Hfq is a member of the (L)Sm protein superfamily and is present among the three domains of life. In bacteria, Hfq assembles as a homohexamer with a donut-like shape containing three distinct RNA binding sites on its surface, the proximal face, the distal face, and the convex circumferential rim of the ring. The proximal face of the ring recognizes uridine-rich sequences, a typical feature of 3′ terminal tails of sRNAs. The distal face preferentially binds adenosine-rich sequences or repeated ARN motifs (composed of an adenosine, a purine, and any nucleotide), a characteristic usually found in mRNAs, allowing the recognition of the sRNA-target regions on the 5′ UTR of mRNAs [17,18]. The rim, which is rich in arginine residues, seems to promote RNA annealing, probably by guiding and facilitating the base pairing between the complementary strands of the sRNA and the target mRNA [19]. The mechanisms underlying the Hfq-mediated sRNA–mRNA interaction are quite diverse. Hfq increases the chance of the RNA molecules to meet by reducing their motility and flexibility; the protein can induce changes in RNA secondary structure by exposing the complementary regions of the RNA molecules, allowing the formation of a stable RNA double strand by base paring; the chaperone increases the local concentration and proximity of the pairs sRNA/mRNA by the specificity of its binding sites [20]. Moreover, Hfq is capable of accommodating multiple RNA molecules and interacting with proteins involved in RNA metabolism and translation, greatly improving the performance of sRNAs as regulators [17]. Hfq is now recognized as a major component of post-transcription regulatory mechanisms in bacterial cells. The output of Hfq-mediated sRNA-mRNA interactions is quite similar to those without mediation, occurring either an activation or a repression of gene expression, with or without mediated degradation by RNAse E [18,21]. Deletion of the gene encoding Hfq in several pathogens like *Salmonella* Typhimurium and *Pseudomonas aeruginosa* has resulted in a strong attenuation of virulence [22]. This evidences the major role played by Hfq-dependent sRNAs in virulence.

A recently characterized RNA chaperone, ProQ, is a RNA-binding protein of the ProQ/FinO domain superfamily, conserved and abundant among α-, β-, and γ-proteobacteria [23]. A study in *Salmonella* Thyphimurium revealed that the sRNAs associated to this chaperone possess extensive secondary structures, distinct from those that associate to Hfq. In addition, only 2% of the identified ProQ-interacting sRNAs also bind to Hfq [24]. The only full description of the mechanism of interaction between ProQ and a sRNA was done by Smirnov et al. [23], who described the ability of ProQ to stabilize the trans-encoded sRNA RaiZ, as well as the ability of the chaperone to mediate the interaction of RaiZ with its mRNA target *hupA*.

## 2. Cystic Fibrosis Lung Infections

Cystic Fibrosis (CF) is the most common and life-limiting autosomal recessive disorder among Caucasians, affecting ~70,000 individuals worldwide [25,26]. The CF disease is characterized by abnormal viscous secretions in several tissues with epithelia, like airways, pancreas, small intestine, liver, reproductive tract and sweat glands. The disease is due to mutations in the gene encoding the Cystic Fibrosis transmembrane conductance regulator (CFTR). Despite the potential damaging effect of *cftr* gene mutations on the affected organs, nowadays the great concern is centered on chronic bacterial airway infections that are responsible for 80 to 90% of mortality [26,27,28]. The expression of CFTR is much more pronounced in the airway’s tissues, where a dysfunctional CFTR causes deficient cAMP-dependent chloride and bicarbonate secretion into airway secretions. Mucins are also secreted to the epithelial surface and then the pH decreases, a factor that impairs the host defense mechanisms. An inefficient inflammatory response, along with a defective mucus clearance, are the main causes of microbial infections in the airways of CF patients [29]. A wide array of bacteria can colonize the airways of CF patients, like *Staphylococcus aureus*, *Haemophilus influenzae*, *Stenotrophomonas maltophilia*, and *Mycobacteria* sp other than *M. tuberculosis*. Fungi like *Aspergillus fumigatus* have also been found in patients’ sputum. All these microbes can contribute to the decline of lung functions [30]. However, the major threats to CF patient’s survival are *P. aeruginosa*, bacteria of the *Burkholderia cepacia* complex (Bcc), and *Achromobacter xylosoxidans.* These bacteria can persist in the airways and provoke chronic infections that are the main cause of morbidity and early mortality of CF patients. *P. aeruginosa* is the leading cause of CF patients infections, being responsible for chronic infections in approximately 80% of CF adults [30,31]. Progresses recently achieved in the development of aggressive early eradication of *P. aeruginosa* led to an enhancement on prognosis [32]. Bacteria of the Bcc are the second major microbial threat to CF patients. Despite their lower incidence (only 3%–4% of CF patients are infected by Bcc), these bacteria are especially feared due to the easy transmission among the patients, the extensive antibiotic resistance, and the risk of cepacia syndrome, a fulminating pneumonia that leads to patient death in a short period of time [33,34].

This review focuses on the current knowledge of sRNAs from the CF major pathogens *P. aeruginosa* and Bcc. Since sRNAs are major regulatory molecules involved in bacteria virulence, we review in this work some sRNAs involved in virulence of *P. aeruginosa*, as well as the present scarce knowledge on Bcc sRNAs and their possible roles on bacteria virulence [35].

## 3. sRNAs of *Pseudomonas aeruginosa*, the Major CF Pathogen

*P. aeruginosa* is a Gram-negative species recovered from a wide range of environmental niches. The species is an opportunistic pathogen to animals and humans and a serious threat to public health [36]. *P. aeruginosa* is responsible for severe infections not only among CF patients, but also in other immunocompromised patients and patients with severe burn injuries [37]. It is still to be fully elucidated how an environmental species has become a major cause of nosocomial infections worldwide. However, it is known that the overuse and misuse of antibiotics have led to the selection of multiple antibiotic resistant strains, against which very few therapeutic options are available [32,36]. In a review, Potron et al. [38] showed that some markers of acquired resistance to antibiotics were found in strains isolated all over the world. The ability of this bacterium to produce a wide range of opportunistic infections and to originate strains resistant to various antibiotics can be attributed to its large genome (typically with more than 6 Mbp), containing a particularly high proportion of regulatory genes, as well as a large number of genes involved in the catabolism, transport, and efflux of organic compounds [32]. Although *P. aeruginosa* infections can be eradicated during acute lung infections in most of the cases, when a chronic infection is established, its eradication is hardly achieved [39]. The most striking examples of infections by *P. aeruginosa* are chronic lung infections in CF patients, which occur in over 60% of adult patients [40]. When invading the CF lung, *P. aeruginosa* has to surpass a heterogeneous, hostile, and stressful environment, being exposed to osmotic stress due to the viscous mucus, oxidative and nitrosative stresses due to host defenses, sublethal concentrations of antibiotics, and the presence of other microorganisms [32].

Acute infections by *P. aeruginosa* are among the best characterized; it is known that the expression of their virulence genes is controlled by extremely complex, interweaving regulatory circuits and multiple signaling systems. To achieve a successful acute infection, *P. aeruginosa* produces an impressive array of virulence factors, such as the Type 3
Secretion Systems (T3SS) to secrete various exotoxins; the quorum sensing (QS) systems to control numerous important secreted components including pyocyanin, elastase, cyanide, and rhamnolipids, exhibiting a motile phenotype [32]. On the other hand, *P. aeruginosa* virulence factors that are important in chronic infections are still to be fully elucidated. It is known that in chronic infections, *P. aeruginosa* strains became less inflammatory and less cytotoxic, and also lose structures responsible for adherence and motility, like the flagellum and pili. Other characteristics include the appearance of mucoidy, due to alginate production, the emergence of highly antibiotic resistant variants, changes in lipopolysaccharide structure including altered lipid A and loss of O-antigen, and alterations in quorum sensing [39]. The GAC system network was pointed as controlling the reversible transition from acute to chronic infections, incorporating the two-component regulatory system GacA/GacS, two other sensor kinases RetS/LadS, the small regulatory protein RsmA, and sRNAs RsmZ and RsmY [32]

To successfully infect their hosts, *P. aeruginosa* must achieve fast and precise regulation of its metabolism. sRNAs are one of the best means to achieve such a fast and tight regulation, and this is most probably why these bacteria possess a so vast array of those posttranscriptional regulators. A total of 573 sRNAs are known to be expressed by the strain PAO1 and 233 sRNAs by the strain PA14 [41]. PAO1 is a moderately virulent strain isolated from a wound, widely used as a reference strain in comparison to the highly virulent isolate PA14, a representative of the most common clonal group found worldwide [41]. 126 sRNAs are common to both strains. However, an in silico study evidenced that the strains share more sRNAs in their genomes than the ones detected [42]. This means that, although some sRNAs are encoded in the genome of both strains their expression could be strain-specific or environmental-dependent [42]. For the PAO1 strains, 149 sRNAs were already annotated and 117 were experimentally validated, the remaining predicted by similarity searching. In the case of the PA14 strains, 136 sRNAs are annotated and 66 were experimentally validated (data obtained from the sRNAs database BSRD [43]).

Although the number of characterized sRNAs is remarkably reduced compared to the total of predicted sRNAs in *P. aeruginosa*, it is still a huge number when compared to other organisms. A boost in the characterization of sRNAs from *P. aeruginosa* strains PAO1 and PA14 took place in recent years. At least one half of the characterized sRNAs were found to be dependent or mediated by the Hfq chaperone (Table 1). Among the sRNAs characterized are the RsmY and RsmZ, both were involved in the switch from the motile to the sessile modes of living, modulating the expression of T3SS and T6SS [44]. However, only the role of RsmZ in cell motility and biofilm formation was proved. Like RsmY and RsmZ, RsmW is also involved in the RsmA regulation, being implied in the regulation of biofilm formation [45,46]. Moreover, ErsA and SrbA are two sRNAs that also play a role in the regulation in biofilm formation. ErsA is part of the response to envelope stress and exerts its regulatory role by modulating the activity of the checkpoint bifunctional enzyme AlgC with phosphomannomutase/phosphoglucomutase activities and expression of the AmrZ regulon [47]. SrbA has also a role in the pathogenesis of *P. aeruginosa* PA14, although the mechanisms underlying such regulation are still to be elucidated [48]. RgsA and NrsZ are both important in swarming motility. Both are regulated by two-component regulatory systems (TCS). NrsZ is regulated by the TCS NtrB/C and perform its role through its involvement in the modulation of *rhlA* expression, and thus controlling the production of rhamnolipid surfactants [49]. RgsA is regulated by the GacS/GacA TCS, having as targets *fis* and *acpP*, leading to the regulation of pyocyanin synthesis [37].

Las, RhL and PQS are the best known *P. aeruginosa* quorum sensing systems. The Pseudomonas quinolone signal (PQS) plays a central role in the quorum sensing network in *P. aeruginosa*, linking the three systems. sRNAs PrrF1/2, ReaL, and PhrS are major players in PQS modulation, all of them playing a role in the expression of virulence genes. PhrS is involved in the regulation of expression of PqsR, a transcriptional regulator of genes involved in PQS and pyocyanin synthesis [50]. ReaL acts by positively regulating *pqsC* expression, in a response modulated by the Las system [51]. PrrF1/2 works as an iron sensor, leading to an interruption of expression of genes encoding iron-binding proteins in low iron conditions. The transcription activator *antR* is also repressed, avoiding the degradation of anthranilate, a precursor of PQS synthesis. [52,53]. The sRNA PrrH that was not linked to *P. aeruginosa* virulence also plays an important role in iron availability, by heme homeostasis [54]. In this context, the CrcZ sRNA also seems to be involved in iron regulation. Like RsmY/Z/W, CrcZ also sequesters proteins, in this case the Hfq chaperone. Although it was initially characterized as a regulator of carbon catabolite repression, recent evidence points out CrcZ as a PrrF regulator, interfering with the Hfq-mediated repression of *antR* [53,55]. Two other sRNAs were also characterized recently, PaiI plays a pivotal role in anaerobic growth conditions and in denitrification, and PesA seems to play a role in *P. aeruginosa* pathogenesis, being involved in Pyocin S3 modulation and resistance to UV radiation [56,57]. Table 1 summarizes the *P. aeruginosa* sRNAs already characterized according to the strain from which they were studied. A full description of the best characterized sRNAs playing a role in *P. aeruginosa* virulence is described below.

### 3.1. RsmY and RsmZ

Chronic infection seems to be the preferred mode of infection by pathogens like *P. aeruginosa*, a premise for its survival and persistence as a population. Since the cellular mechanisms involved in acute or chronic infection are different, the cell is equipped with a complex regulatory network to switch its state. In the case of *P. aeruginosa*, the Csr/Rsm system plays a major role in this switch [52]. This system controls a large variety of physiological processes, like carbon metabolism, virulence, motility, quorum sensing, siderophore production, and stress responses [45]. The system comprises a major translational regulator, the protein RsmA, which negatively regulates mRNA targets by binding to sites containing critical GGA motifs present in the 5′-UTR of mRNAs targets [45] This binding represses the translation of regulons necessary for establishing chronic infections, as the T6SS, quorum sensing systems, exopolysaccharide production, biofilm formation, and iron homeostasis [70,71]. On the other hand, RsmA exerts a positive and indirect regulation of the mechanisms linked to acute infection, like the expression of genes associated with surface motility, T3SS, type IV pili, as well as systems that operate through the cAMP/virulence factor regulator (Vfr) route [72]. RsmA appears to control T3SS gene expression by increasing *exsA* translation through an undetermined mechanism. ExsA is the master transcription factor of all T3SS genes [73]. So, RsmA regulates the switching from planktonic (acute infection) to biofilm (chronic infection) phenotype. RsmA activity is negatively regulated by the two-component regulatory system GacA/GacS that induces the expression of sRNAs antagonist of RsmA, including the RsmY and RsmZ in *P. aeruginosa* (Figure 1) [74]. These two last sRNAs have a secondary structure with several unpaired GGA motifs that sequester RsmA proteins, preventing biding to their targets, and thus enhancing the expression of specific RsmA regulons [45].

Although GacA is a major regulator of *rsmY* and *rsmZ* expression, these sRNAs can also be regulated by alternative regulators, and despite the redundancy, they can have different regulators. It was observed in *P. aeruginosa* that *rsmY* transcription steadly increases during cell growth, whereas *rsmZ* is induced harshly during the late exponential phase of growth [45]. Using swarming motility as a model, Jean-Pierre et al. [61] showed that these sRNAs are differentially regulated depending on the selected growth conditions (planktonic versus surface–grown cells), and that *rsmZ* regulation does not implicate the response regulator GacA in swarming cells. Furthermore, these authors observed that RsmY/Z expression influences swarming motility via the protein HptB, which acts as a negative regulator of these sRNAs and that they do not strictly converge to RsmA [61]. Before a biofilm can be formed, RsmZ is eliminated by the action of ribonuclease G, activated by the TCS BfiSR [45]. Other mechanisms of regulation of RsmY and RsmZ have been discovered recently. AlgR, a TCS response regulator important for *P. aeruginosa* pathogenesis in both acute and chronic infections, seems to affect the antagonizing action of RsmY and RsmZ on RsmA [75]. SuhB regulates the motile–sessile switch in *P. aeruginosa* through the Gac/Rsm pathway and c-di-GMP signaling, regulating multiple virulence factors like the T3SS, swimming motility, T6SS, and biofilm formation [44]. Li et al. [44] reported that this regulation is mediated by the GacA-RsmY/Z-RsmA. MgtE, a magnesium transporter, was demonstrated to play a role in the inhibition of the T3SS transcriptional activator ExsA. The negative regulation is mediated by expression of RsmY and RsmZ, most likely though the TCS GacA/GacS [73]. The polynucleotide phosphorylase (PNPase), which is involved in the regulation of multiple virulence factors like T3SS, T6SS and pili biosynthesis, plays an important role in RsmY/Z stability. This stabilizing ability is due to the PNPase KH-S1 domains that bind to sRNAs stabilizing them, providing a way of pathogenicity regulation by PNPase [76]. Recently, Miller et al. [45] characterized a new sRNA in *P. aeruginosa*, the RsmW. Unlike RsmY and RsmZ, this sRNA is not transcriptionally activated by GacA. RsmW was shown to be upregulated in nutrient-limiting conditions, biofilms, and at higher temperatures [45].

### 3.2. ReaL

ReaL is a highly conserved *P. aeruginosa* sRNA recently characterized by Carloni et al. [51]. ReaL was shown to be a relevant element for *P. aeruginosa* pathogenesis. In the infection model *Galleria mellonella*, deletion of the ReaL encoding gene impaired *P. aeruginosa* PA14 virulence, while the sRNA overexpression resulted in a hypervirulent phenotype. ReaL is involved in the quorum sensing regulatory network architecture, interlinking the Las and PQS QS systems. Both Las and PQS are part of a major network played at least by two additional systems, the integrated quorum sensing system (IQS) and the Rhl QS system [77]. This entire network is closely related with the expression of virulence factors in *P. aeruginosa*. The Las system is composed by the transcriptional regulator LasR, its cognate autoinducer molecule N-3-oxo-dodecanoyl-homoserine lactone (3-oxo-C12-HSL), and the 3-oxo-C12-HSL synthase LasI [51]. PQS, positively regulated by the Las QS system, is an essential mediator of the shaping of the population structure of *P. aeruginosa* and of its responses and survival in hostile environmental conditions [78,79]. PQS synthesis proceeds through the condensation of an anthranilate molecule with a fatty acid to produce the 2-heptyl-4-quinolone (HHQ), which is then hydroxylated by the PqsH enzyme, whose expression is positively regulated by the LasR3-oxo-C12HSL to produce PQS [77,80]. It is also known that ReaL plays a role in the interconnection between both systems. Actually, ReaL is negatively regulated by LasR, and because it is a positive post-transcriptional regulator of the *pqsC* gene, a decrease of PQS will occur. ReaL seems to contribute to the fine co-modulation of HHQ/PQS synthesis [51]. The PQS system plays a vital role in biofilm formation and production of virulence factors such as pyocyanin, elastase, PA-IL lectin, and rhamnolipids [77]. Therefore, ReaL expression triggers the production of some virulence factors, like biofilm formation and pyocyanin synthesis, by a Las-independent way. In addition, ReaL plays a negative role in swarming motility. The upregulation of ReaL expression is observed in the conditions found when colonizing the human lung, like a temperature of 37 °C and low oxygen concentration [51].

### 3.3. ErsA

ErsA is the first sRNA from *P. aeruginosa* that appears to be embedded in the envelope stress response, a critical transduction pathway that impacts pathogenesis in *P. aeruginosa*. ErsA expression depends of the envelope stress responsive sigma factor σ^22^, the major player in the envelope stress-signaling pathway like σ^E^ in *S.* Typhimurium and other enterobacteria. ErsA directly exerts a negative post-transcriptional regulation on the virulence associated *algC* gene encoding the bifunctional enzyme AlgC [68]. AlgC plays a central role in exopolysaccharide biosynthesis, generating the sugar precursors mannose 1-phosphate and glucose 1-phosphate, necessary for the synthesis of polysaccharides like alginate, Pel, Psl, LPS, and rhamnolipids, key components of the biofilm matrix [81]. A fine-tuned regulation exerted by ErsA on AlgC is thought to influence the dynamics of exopolysaccharide biosynthesis, underlying the development of biofilm matrix. This regulation is crucial when the bacterial cell is under envelope stress. The repressive role of ErsA on *algC* expression occurs mainly at the translational level by base pairing with the RBS in a Hfq dependent way [68]. Furthermore, Zhang et al. [82] found that ErsA represses the expression of the OprD porin, the major channel for the uptake of basic amino acids, peptides, and the carbapenem antibiotic, evidencing a role played by this sRNA in antibiotic resistance.

### 3.4. PrrF

PrrF sRNAs are encoded in *P. aeruginosa* genome by two highly homologous genes, *prrf1* and *prrf2*. These sRNAs are encoded in tandem sequences, separated by 95 nt, a characteristic exclusive of this species. The entire *prrF* region also encodes another sRNA, PrrH, transcribed from the 5′ end of *prrf1* to the 3′ end of *prrf2* [83]. Infection experiments in a murine model using a deletion mutant on the *prrF* locus showed that this genetic locus plays a pivotal role in *P. aeruginosa* virulence in acute murine lung infection. The encoded sRNAs are also important for iron and heme homeostasis [64]. PrrH seems to play a role in heme homeostasis and on the expression of virulence factors. However, PrrH is not vital for acute murine lung infection, as shown by Reinhart et al. [54]. Iron acquisition is essential for *P. aeruginosa* virulence and biofilm formation, highlighting the role of regulatory systems on the maintenance of iron homeostasis, with this sRNA playing a critical role in bacteria survival in the host [64]. PrrF is responsible for the iron-sparing response, an effect observed under conditions of iron limitation. PrrF exerts its regulatory activity by repressing mRNAs encoding iron-containing proteins like the iron cofactored superoxide dismutase SodB, a putative bacterioferritin, a heme-cofactored katalase and succinate dehydrogenase. Iron “sparing” is essential when the intracellular iron concentration is low [54]. Although iron is an essential metallonutrient, when it accumulates it becomes toxic, due to its ability to catalyze the formation of reactive oxygen species via Fenton reactions. The regulation of such a homeostasis is mediated by the ferric uptake repressor protein Fur [84]. Fur has the ability to bind to iron in iron-enriched environments, becoming an active repressor of PrrFs, leading to a more extensive use of iron by the proteins otherwise downregulated by PrrFs [64]. PrrF sRNAs also repress the expression of AntR, a transcription activator of genes involved in the conversion of anthranilate to catechol. By this way, the anthranilate is channeled into the biosynthetic pathway of the PQS, a system that leads to the expression of several virulence factors as already mentioned above [64,85]. Furthermore, Sonnleitner et al. [53] added another sRNA to the already extensive armory of *P. aeruginosa*, the CrcZ sRNA, which impairs the binding of PrrFs to *antR*. Since the riboregulation played by PrrFs is mediated by Hfq, the CrcZ functions like a sponge, binding to Hfq with a higher affinity compared to *antR*, inhibiting its binding. This mechanism leads to the de-repression of *antR*.

### 3.5. PesA

PesA, characterized by Ferrara et al. [57], is encoded in the pathogenicity island PAPI-1, being expressed by the highly virulent strain PA14, as well as by several clinical isolates obtained from CF patients. The *P. aeruginosa* strain most commonly used for research, PAO1, does not encode PesA. The sRNA is expressed under temperature and oxygenation conditions resembling those found in the CF lung, namely 37 °C and low oxygen. PesA was shown to be important for *P. aeruginosa* pathogenicity in CF bronchial cells. PesA is involved in the post-transcriptional regulation of genes involved in S-type pyocin production, like pyocin S3. Pyocin S3, like other soluble pyocins, has a killing activity that causes cell death by DNA cleavage. Pyocins are produced by more than 90% of the *P. aeruginosa* strains, helping *P. aeruginosa* in niche establishment, protecting the bacteria from competition [57,86]. By deleting *pesA*, an increased susceptibility to UV irradiation and to the fluoroquinolone antibiotic ciprofloxacin was observed, correlated with a decreased level of pyocin expression. A similar effect was observed when deleting the gene encoding pyocin S3. Although contradictory, PesA seems to play an important role in the response to DNA damage by promoting the synthesis of pyocin S3 [57]. The mechanism underlying this response is not yet clear.

## 4. The *Burkholderia cepacia* Complex and the Emerging Knowledge on Its sRNAs

When compared to *P. aeruginosa*, respiratory infections caused by Bcc species have a low prevalence on CF patients. However, bacteria of this complex are still one of the most feared pathogens and represent a higher risk for those patients.

*Burkholderia cepacia* complex (Bcc) is a group of opportunistic human pathogens presently composed of 23 closely-related species [87]. Apart from being a major threat for CF patients, Bcc bacteria can also cause lethal infections among chronic granulomatous disease (CGD) patients, immunocompromised patients (e.g., HIV infected), cancer patients, and other chronic patients [88]. *B. cenocepacia* and *B. multivorans* remain the predominant Bcc species worldwide that are responsible for infections in CF patients [89,90]. Bcc species possess genomes larger than those of *P. aeruginosa* strains, usually with a length of more than 7 Mbp. This large genome is thought to confer the bacterium an enormous ability to overcome antimicrobial therapies, as well as the host immune response, being in constant evolution inside the host lung [91].

The defective mucus clearance of the CF lung is the perfect niche for a persistent infection by *P. aeruginosa*, as well as by Bcc bacteria that tends to occur after *P. aeruginosa* colonization, usually superseding it [92,93]. Moreover, Bcc infections are easily propagated among CF patients. Bcc bacteria possess an extraordinary resistance to antibiotics [94,95]. About twenty percent of CF patients infected with Bcc bacteria develop the cepacia syndrome, a necrotizing pneumonia, often accompanied by septicemia, leading to a rapid, irreversible and deadly decline of the respiratory function [96,97].

Bcc virulence resembles several traits of *P. aeruginosa* virulence, due both to the fragility of the host and to the bacterium extraordinary evolutionary development. Once the colonization of the respiratory tract is established and the bacteria are attached to epithelial cells, they have to overcome a strong immunological response [98]. The successful invasion of host internal system is achieved by mechanisms of penetration like paracytosis and invasion as a biofilm [99,100]. Among the virulence factors expressed by Bcc species the extracellular lipase, metalloproteases and serine proteases, and some structures of the bacterial surface such as flagella, pili, and the lipopolysaccharide (LPS) can be highlighted [96,100]. Exopolysaccharide (EPS) production, namely Cepacian, also represents a major virulence factor, important for biofilm formation and to protect Bcc from the host defense machinery, as well as from antimicrobial therapy [101,102,103,104].

Like in *P. aeruginosa*, several noncoding RNAs have been associated with regulatory mechanisms involved in Bcc virulence, as is the case of the modulation of the adaptation to the host changing environment, the adherence to, and invasion of host cells, as well as the replication. However, the Bcc noncoding virulome (sRNAs playing a role in virulence) is still poorly understood compared with *P. aeruginosa*.

### 4.1. Discovering Bcc Noncoding Transcriptome

A pioneer systematic search for sRNAs in Bcc was first described by Coenye et al. in 2007 [105], who predicted an array of putative noncoding RNAs (ncRNAs) from *B. cenocepacia* J2315, using *R. solanacearum* GMI1000 genome as a reference. Two-hundred-and-thirteen ncRNA genes, ranging in size from 53 to 1243 nt, were predicted and described. The expression of only four of these sRNAs was confirmed by microarray experiments using total RNA from cells grown on a 10% (*w*/*v*) CF sputum [105]. The expression of one sRNA has confirmed by real-time PCR; none of these putative sRNAs were functionally characterized.

In order to understand the mechanisms underlying the early stage of Bcc infections, Drevinek et al. [106] performed a microarray-based transcriptomic analysis of *B. cenocepacia* grown from sputum recovered from CF patients. The strain was found out to be indistinguishable from *B. cenocepacia* J2315, a strain involved in multiple fatalities. The genes found to be upregulated were mainly involved in antibiotic resistance, the antioxidant response to reactive species, iron metabolism, and virulence factors such as flagella and metalloproteases. A thorough analysis of transcripts originating from intergenic regions was carried out, leading to the identification of 88 upregulated and 126 downregulated sequences. However, no further characterization of these regions was performed to identify putative sRNAs.

In 2009, Yoder-Himes et al. [107] used Illumina RNA-seq techniques to understand the mechanisms of transcriptional regulation associated to *B. cenocepacia* environmental adaptation. Two strains, one isolated from soil (*B. cenocepacia* HI2424) and another from a CF patient (*B. cenocepacia* AU 1054), were grown under conditions mimicking each environment, and their transcriptomes were analyzed. In both conditions a higher number of genes was transcribed in strain HI2424 than in strain AU 1054. When grown under CF-like conditions, the strains exhibited differences in the expression of genes mainly from chromosome 1, while growth under soil conditions led to differences in the expression of genes mainly encoded in chromosomes 2 and 3. Regarding sRNAs, 13 were identified (ncRNA1 to ncRNA13), but only one seems to be induced under CF-mimicking conditions. Those sRNAs have been predicted to be highly structured. The expression of 4 sRNAs was confirmed by northern blot, allowing the authors to assume that the sequences were indeed expressed in vivo. The majority of these molecules are conserved among the *B. cenocepacia* strains and the Bcc (Appendix A).

Effective response to reactive oxygen species by bacteria is an important feature regarding the adoption of disinfection measures, as well as during exposure to oxidative burst by the host defenses. To evaluate the response to reactive oxygen species (ROS), Peeters et al. [108] used microarray transcriptomic analysis and qPCR to analyze whole cell responses of *B. cenocepacia* J2315 sessile cells exposed to H_2_O_2_ and NaOCl. The profile of genes up- and downregulated was found to be similar to that of planktonic *P. aeruginosa* PAO1 cells exposed to similar conditions. Many genes involved in prevention (counteracting or repairing of the damage from oxidative stress) were found to be upregulated, as well as some genes involved in the synthesis and assembly of flagella. Moreover, several transcripts from intergenic regions were found, 39 and 56 were upregulated upon exposure to H_2_O_2_ and NaOCl, respectively, whereas 54 and 68 were found to be downregulated. To avoid false positives, the authors selected as putative noncoding RNAs the regions whose expression pattern is different from the flanking genes. Eleven and 20 intergenic regions were selected as putative noncoding RNAs upregulated with H_2_O_2_ and NaOCl, respectively. Transcripts from intergenic regions IG1_2935724 and IG1_3008003 were upregulated in both conditions and their expression was confirmed by qPCR. The ncRNA4 and ncRNA6, found by Yoder-Himes et al. [107], are located in those intergenic regions, thus confirming their expression. A match was found between the IG1_2935724 and the secondary structure of 6S RNA consensus structure. Seven out of the 11 sRNAs upregulated upon exposure to H_2_O_2_ were also found significantly altered in *B. cenocepacia* J2315 cells of CF sputum [106].

Coenye et al. [109] used microarrays to study transcriptomic differences between planktonic and sessile cells of *B. cenocapacia* J2315 exposed to chlorhexidine. The authors found that sessile cells are more resistant to chlorhexidine (0.015%) than planktonic cells. Furthermore, sessile cells highly expressed genes encoding efflux systems related to drug resistance, as well as membrane-associated proteins and regulators. The results suggest that sessile cells engaged a global expression program to evade the biofilm since an adhesin was downregulated and genes encoding chemotaxis and motility-related proteins were upregulated. This study also allowed the identification of several intergenic regions upregulated in the presence of chlorhexidine. After discarding the regions with an expression level similar to the closest coding sequences (CDS) and the ones whose probes overlapped the coding sequences, 19 intergenic regions were suggested as putative sRNAs. As presented in Appendix A, these regions are largely conserved among *B. cenocepacia* strains, less conserved among the Bcc members, and only one of them is conserved in the *B. pseudomallei* group. The majority of the predicted sRNAs seems to have a stable secondary structure, however no significant match was found in the Rfam database [109].

Previous work from our research group led the experimental identification of 24 sRNAs from *B. cenocepacia*, based on co-purification of total RNA and the chaperone Hfq [110]. Expression of these sRNAs was confirmed by northern blot and characterized in silico. The only consistent results were related to Bc7, for which the *hemB* was predicted as mRNA target, and the *Salmonella* Paratyphi RybB as homolog. In addition a cis-encoded sRNA, named h2cR, was also previously reported by our research group as being involved in the regulation of the *hfq2* gene that encodes an Hfq-like chaperone in Bcc [111].

The genome of *B. cenocepacia* KC-01, a strain isolated from the coastal saline soil, was recently sequenced by Ghosh et al. [112]. Several potential sRNAs were identified by in silico analysis of the genome, and the expression of seven putative sRNAs was confirmed (Bc_KC_sr1-7). Bc_KC_sr1 and Bc_KC_sr2 were upregulated in response to iron depletion by 2,2′-bipyridyl. Bc_KC_sr3 and Bc_KC_sr4 were induced under the presence of 60 µM H_2_O_2_ in the culture medium. Alterations on the temperature and incubation time also induced the expression of Bc_KC_sr2, 3, and 4. Searches within the RFAM and BSRD databases led to the identification of candidate738 of *B. pseudomallei* D286, tmRNA and 6S RNA as homologs of Bc_Kc_sr4, 5, and 6, respectively. Interestingly, this group of sRNAs is extensively conserved among members of the Bcc and *B. pseudomallei* groups (Appendix A). Several targets were predicted for these sRNAs, like Fe-S cluster and siderophore biosynthesis, ROS homeostasis, porins, transcription, and translation regulators.

### 4.2. A Deeper Approach: Bcc sRNAs Expressed under Biofilm Formation Conditions

Based on the importance of biofilm formation to antibiotic resistance, Sass et al. [113] analyzed the transcriptome of *B. cenocepacia* J2315 grown in biofilms by differential RNA-sequencing (dRNA-seq). dRNA-seq differs from RNA-seq on a selective sequencing of primary transcripts. The techniques, used for the first time for a Bcc organism, allow mapping of the transcription start sites (TSS) based on the difference of primary and processed transcripts ends. Primary transcripts carry a 5′ triphosphate end, while processed transcripts like tRNA and rRNA, carry a 5′ monophosphate. Using the 5′ P-dependent terminator exonuclease (TEX) to degrade the 5′ monophosphate, it is possible to distinguish between primary and processed transcripts [114]. As a complementary approach, Sass et al. also carried out a global RNA-seq (gRNA-seq) to obtain a large coverage of the whole transcript length and the 3′ end of transcripts, usually lost by dRNA-seq of longer transcripts. The additional use of 5′ RACE allowed the authors to end up with 2089 genes annotated as expressed under biofilm conditions and to identify alternative start codons for some genes and novel protein sequences. A total nine sRNAs with homologs present in the Rfam database and other seven match putative sRNAs that were experimentally identified in previous studies. The sRNAs with homologs in the Rfam database were the 6S RNA; two phage-related regulatory RNAs located on genomic island BcenGI9; two conserved regulatory motifs; the SAH riboswitch located upstream of BCAL0145; an adenosylhomocysteinase; the *sucA* RNA motif located upstream of *sucA* (BCAL1515), an enzyme of the citric acid cycle; and four sRNAs from the family named “toxic small RNAs”, whose expression in *B. cenocepacia* have already been confirmed by northern blot but with unknown functions. From the transcript sequences containing a TSS in intergenic regions that did not match with a gene and without a hit in the Rfam database, the authors selected only those obeying to the following criteria: strong transcription initiation with a coverage >300 reads in dRNA-Seq data, a defined 3′ end in dRNA-Seq data or a transcript appearing short (<500 nt), and truncated or missing in gRNA-Seq data. Upon selection, the sequences were compared with the experimentally confirmed sRNAs already published by other authors. Six putative sRNAs matched those found by Yoder-Himes et al. [107], five of them upregulated on soil conditions and one on cystic fibrosis sputum. Another transcript matched one sRNA previously identified by using co-purification with Hfq [110]. The expression of these sRNAs in other studies, together with their overexpression under conditions of biofilm formation, suggests a role for these sRNAs on biofilm regulation.

Sass et al. [115] refined their analysis on putative sRNAs found in association with biofilm formation. From a total of 148 TSS found in intergenic regions, 41 transcripts were classified as rho-independent (RIT), and 82 transcripts as derived from 5′ UTR of the downstream gene. Fifteen sRNAs were selected based on the following criteria, the number of starts at TSS (≥250), the z-score (<−1), and the conservation on Bcc (≥13 strains). 3′RACE and Northern blot analysis were used to confirm the length of sRNAs. The expression of 14 sRNAs was quantified by qPCR in different experimental conditions, all compared with planktonic cultures. Twelve sRNAs were upregulated in biofilms. Some sRNAs were slightly upregulated by osmotic or pH stress. Nutrient starvation also induced the expression of 12 sRNAs, as well as the expression of Hfq chaperone. Five of these sRNAs were also more expressed under glucose-rich medium.

Further work performed by these authors led to the identification of ncS63 as highly expressed under conditions of low-iron. This sRNA is located upstream of BCAL2297, which encodes for the hemin-uptake protein HemP, whose expression is under the regulatory control of the Fur repressor. The predicted targets of ncS63 are related to iron homeostasis like *sdhA*, a target of RyhB from *E. coli*. RyhB is a sRNA involved in iron homeostasis regulated by the Fur repressor, and a possible functional analogue of ncS63. Recently, RyhB was hypothesized to play an important role as a mediator of bacterial resistance to multiple antibiotics and stresses [116]. Further investigation is needed to check if ncS63 is a RyhB functional homolog. The overall prediction of sRNAs targets using CopraRNA and RNApredator led to similar clues about the mechanisms that can be under regulation of these sRNAs, like transcriptional regulators, carbon compound transport and metabolism, and cell envelope components such as outer membrane proteins and porins. The transcriptional regulator BCAL1948 has been predicted to be a target of nine sRNAs, exhibiting particularly extensive complementarity to ncS11. sRNAs containing double-hairpin were predicted to target genes involved on the metabolism and transport of amino acids, carbohydrates, and aromatic compounds. The ncS16 has several hits of genes related to outer membrane and cell envelope components, and genes for transport of inorganic compounds, whereas the best hits for ncS63 were genes involved in energy production and detoxication of reactive oxygen species. Regarding the putative sRNAs derived from 5′UTRs, most likely they have a cis-regulatory function, since some 3′ ends of adjacent genes are homologs of genes that possess cis-regulatory structures in other species. A possible trans-regulatory function remains to be investigated.

### 4.3. ncS35, a Functionally Characterized B. cenocepacia sRNA

While in other organisms sRNAs are being characterized for decades, their characterization in Bcc is starting. ncS35 was identified by Sass et al. [115] in the course of *B. cenocepacia* J2315 biofilm transcriptomic analysis, being the first trans-encoded characterized sRNA from *B. cenocepacia*. The expression of ncS35 was found to be upregulated in cells grown as biofilm and in minimal medium compared to planktonic cells grown in rich medium. Increased aggregation, higher cell metabolic activity, higher growth rate, and increased susceptibility to tobramycin were described for deletion mutant cells. Moreover, an upregulation of the phenylacetic acid and tryptophan degradation pathways was observed in the mutant cells, and the first gene of the tryptophan degradation pathway was predicted to be a putative target of ncS35. This sRNA seems to lead to an attenuation of the metabolic and growth rates, which can be a way of cells to protect themselves against stress conditions. A slow growth rate is observed on *P. aeruginosa* biofilms in CF patient’s sputum, and it is also known that bacteria on slow growth rate, even in planktonic cultures, have an increased resistance to antibiotics, including tobramycin [117,118]. Slow growth rates and metabolic activity are characteristics of persister cells and probably the cause of drug resistance. Since the effect of antibiotics is mainly due to the inhibitory action of some metabolic pathways, slow-growing cells are somehow protected from antibiotics and are more likely to develop drug resistance [119]. The occurrence of persister cells in Bcc infections is known, and it has also been observed in *B. cenocepacia* J2315 after treatment with tobramycin [120]. A better characterization is required to fully understand the roles of ncS35 on *B. cenocepacia* pathogenicity, antibiotic resistance, and possibly on the formation of persister cells.

### 4.4. Compiling the Bcc Predicted sRNAs

Although a few sRNAs from Bcc organisms are functionally characterized, a large number of putative sRNAs have been described by several research groups. This information was gathered, and we present on Appendix A all predicted and confirmed sRNAs, as well as the intergenic regions, which can include candidate noncoding RNAs described so far for *B. cenocepacia*. Predicted sRNAs for which data is somehow confusing and uncertain, or the methods applied were not accurate enough, were not considered. *B. cenocepacia* was chosen because it is the one of the Bcc species with the noncoding transcriptome better characterized. A total of 167 putative sRNAs were included on Appendix A, seven from *B. cenocepacia* KC-01, 13 from *B. cenocepacia* AU 1054, and 147 from *B. cenocepacia* J2315, the majority of them identified by Sass et al. [115]. Putative sRNAs are unevenly distributed throughout the three chromosomes: 67.5% are located on chromosome 1, 28.1% on chromosome 2, and 4.4% on chromosome 3. The length of the replicons on the J2315 strain is 3,870,082, 3,217,062, and 875,977 bp, respectively [121]. The chromosomes 1 and 2 are approximately equally sized; however, chromosome 1 accommodates twice the sRNAs compared with the second chromosome. This suggests that chromosome 1 sRNAs are more relevant for core functions of bacterial metabolism (e.g., cell division). On the other hand, since the regulation of the other two replicons encode genes more related to accessory functions, the regulatory noncoding RNAs encoded in those replicons are probably expressed in response to specific conditions that may have not been assessed yet [121]. Wong et al. [122] used a transposon mutant pool and identified 383 possible essential genes on J2315 strain, 90% of which was present on chromosome 1. Such a strategy can not only enable the discovery of crucial coding genes expressed on different conditions, but might also unveil the Bcc noncoding transcriptome, allowing the discovery of important genes related to some features like virulence and antibiotic resistance.

Regarding the sequence conservation, 93% of the putative sRNAs listed in Appendix A are conserved or semiconserved among the species, 70% among the Bcc bacteria and 24% also in the *B. pseudomallei* group. Although the *B. pseudomallei* group is phylogenetically distant and those bacteria display a distinct pathogenicity and epidemiology, a high level of sequence homology is found in almost 25% of the putative sRNAs identified so far [123]. It is also interesting to access the conservation of the putative sRNAs in each chromosome, since a higher conservation can be observed on chromosome 1, much more extensive than chromosome 2 and 3. A huge difference is observed on the conservation of sRNAs on the Pseudomallei group, with a conservation (and semiconservation) of ~25% on sRNAs from chromosome 1, whereas sRNAs from chromosome 2 and 3 exhibited less than 4% conservation. This is also evidence that Bcc comprises more closely related species than the Pseudomallei group, mainly regarding the accessory functions.

Several sRNAs were identified by more than one experimental approach and given different names. For instance, ncRNA4, ncS17, and Bc_KC_sr6 are the same sRNA, which was found on the intergenic region IG1_2935724. This sRNA shares homology with the 6S RNA, a regulator of RNA polymerase (RNAP) that is widespread among members of the Bacteria domain. This sRNA was found as expressed in distinct situations like soil conditions, biofilm formation and response to ROS [124]. Moreover, an upregulation of the 6S RNA was found in response to oxidative stress, further implying it with a broader role than just on core metabolism, such as on the protection of the cell in the host environment [108]. The possible role of 6S RNA in pathogenesis or in host survival has been described for several bacteria [124]. The homologs ncRNA6 and ncRI12 are both located on the intergenic region IG1_3008003 and are probably the same sRNA. This sRNA was overexpressed during biofilm formation and upregulated under soil conditions, exposure to chlorhexidine, and ROS, suggesting a role on protection against stress conditions. ncS06, homolog of ncRNA7, was found to be expressed under soil conditions and biofilm formation. Prediction of ncS06 mRNA targets revealed several peptidase-encoding genes [115]. ncS11 is a homolog of ncRNA13 and was found to be upregulated on soil conditions, as well as on biofilms, rich medium and starvation conditions [115]. ncS11 has an extensive complementarity to BCAL1948, whose expression was found to be downregulated under biofilm formation. BCAL1948 is a protein of the LysR-type transcriptional regulator (LTTR) family, whose global transcriptional regulators can be found in diverse organisms. The regulated genes are hypothesized to be involved in a wide range of processes like cell division, metabolism, nitrogen fixation, oxidative stress responses, motility, quorum sensing, attachment, secretion, virulence, and toxin production [125].

## 5. Conclusions

There is a clear contrast between the knowledge currently available about sRNAs characterization, regulatory pathways and interactome in *P. aeruginosa* and Bcc bacteria. In *P. aeruginosa*, fourteen sRNAs have already been functional characterized. Some *P. aeruginosa* sRNAs are involved in the pathogenicity and virulence of the organism, like biofilm formation, iron metabolism, quorum sensing regulation, response to stress, and host cell invasion. On the other hand, in Bcc bacteria the first biological function of a sRNA is now being described. This sRNA is ncS35, a growth regulator that seems to do not interfere with the virulence of Bcc bacteria. Despite the scarce characterization of sRNAs in Bcc, over the last 10 years at least 167 putative sRNAs were identified in *B. cenocepacia* as being expressed on CF sputum, biofilm formation, response to oxidative stress, among others specific conditions.

In the peculiar environment of the CF lungs, it is interesting to note that both *P. aeruginosa* and Bcc bacteria can cause infections presenting a high degree of intrinsic and acquired resistance to antibiotics. Both opportunistic pathogens have two of the largest genomes known among prokaryotes, which were predicted to encode a huge number of sRNAs. The functional characterization of these sRNAs is only beginning in those CF pathogens. However, the already established or predicted roles of some sRNAs in modulating cellular processes linked to pathogenesis, suggests their yet unexplored importance for the physiology and pathology of these pathogens, as well as their likely influence in the outcome of *P. aeruginosa* and Bcc infections. The assessment of the functional roles of sRNAs in the host–pathogen interaction is expected to provide additional fundamental knowledge for the development of next-generation antibiotics inspired by sRNAs and their targets.

## Figures and Tables

**Figure 1 ijms-19-03759-f001:**
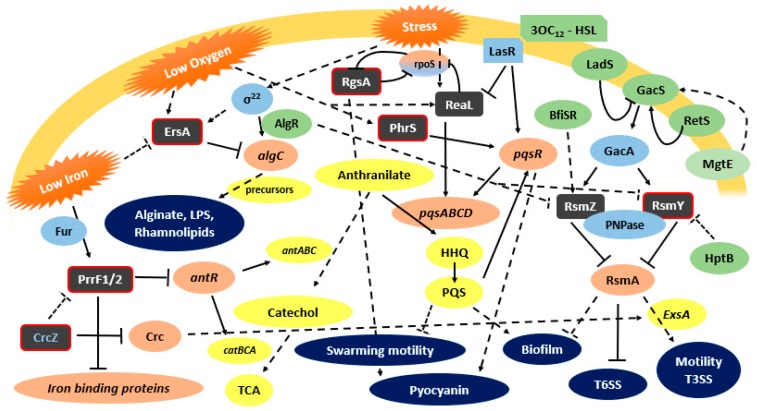
*P. aeruginosa* functionally characterized sRNAs and their involvement on virulence regulatory pathways. Grey box—characterized sRNAs (with red border- Hfq biding); Green box—indirect regulator, Light Blue box—direct regulator; Coral box—direct target; Yellow box—indirect target; Dark Blue—virulence factors; ↓: positive control; ┴: negative control; dashed lines: indirect control.

**Table 1 ijms-19-03759-t001:** Genome location and regulatory interactions of the sRNAs already characterized in *P. aeruginosa*.

Strain	sRNA	Annotation	Length	Genomic Location	Strand	Category ^1^	RBP	Function/ Pathway	Regulators		Targets	Source
									Direct	Indirect		
PAO1	RsmZ	spae4058.1	119	4057542-4057660	Rv	trans	ND	Cell motility Biofilm formation T3SS-T6SS switch	GacS/GacA PNPase	HptB; LadS; RetS; AlgR; BfiSR	RsmA	[44,46,58,59,60]
PAO1	RsmY	spae587.1	124	586867-586990	Fw	trans	Hfq binding	T3SS-T6SS switch	GacS/GacA PNPase	HptB; LadS; RetS; AlgR;	RsmA	[44]
PA14	RsmY	PA14_06875	124	596840-596963	Fw	trans	Hfq binding	Cell motility	GacS/GacA	HptB; LadS; RetS; AlgR;	RsmA	[59,61]
PAO1	PhrS	spae3706.1	213	3705309-3705521	Rv	trans	Hfq binding	PQS regulation Virulence gene regulation	ANR		*pqsR*	[50,62]
PAO1	PrrF1	spae5284.1	152	5283960-5284111	Fw	trans	Hfq binding	Iron acquisition and storage PQS regulation Virulence gene regulation	Fur		*antR; sodB; PA4880; acnB; m-acnB; sdhD*	[52,63,64]
PAO1	PrrH	spae5284.2	325	5283995-5284319	Fw	trans	ND	Heme homeostasis	Fur		*acnB;* *m-acnB; sdhD; nirL*	[52,54]
PAO1	PrrF2	spae5285.1	149	5284172-5284320	Fw	trans	Hfq binding	Iron acquisition and storage PQS regulation Virulence gene regulation	Fur		*antR; sodB;* *PA4880; acnB;* *m-acnB; sdhD*	[52,63,64]
PAO1	CrcZ	spae5309.3	407	5308587-5308993	Fw	trans	* Hfq binding	Carbon catabolite repression	CbrA/B		Hfq; Crc	[55,65,66]
PAO1	RgsA	spae3319.1	197	3318663-3318859	Fw	trans	Hfq binding	Swarming Motility Virulence	RpoS	GacS/A	*fis; acpP;* *rpoS*	[37,67]
PAO1	NrsZ	PA5125.1	226	5775397-5775623	Fw	trans	ND	Swarming Motility		NtrB/C	*rhlA*	[49]
PAO1	ErsA	spae6184.2	201	6183500-6183700	Rv	cis	Hfq binding	Envelope stress response Biofilm formation		σ^22^	*algC; oprD;* AmrZ regulon	[47,68]
PA14	ErsA	spau6457.2	201	6456400-6456600	Rv	cis	ND	Envelope stress response		σ^22^	*algC*	[68]
PA14	PaiI	PA14_13970.1	126	1198928-1199053	Rv	trans	Hfq binding	Anaerobic Growth Denitrification		NarXL		[56]
PA14	SrbA	PA14_30065	239	2604298-2604536	Rv	trans	ND	Biofilm formation Virulence				[48]
PA14	ReaL	spau1600.1	201	1599900-1600100	Rv	trans	ND	PQS synthesis Regulation of virulence	LasR-3OC_12_HSL RpoS		*pqsC* *rpoS*	[51,69]
PA14	PesA	spau5289.2	401	5288100-5288500	Fw	cis	ND	Pyocin S3 modulation Resistance to UV radiation Virulence				[57]

Abbreviations: ANR - Anaerobic transcriptional regulator ANR; RBP–RNA Binding Protein; Rv–reverse; Fw–forward; ND–Non-defined; * Hfq binding and sequestering. ^1^ chromosome location related to the mRNA target.

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
