# Peer review of "Small Noncoding Regulatory RNAs from Pseudomonas aeruginosa and Burkholderia cepacia Complex"

_ijms, 2018, doi:10.3390/ijms19123759_

Round 1

Reviewer 1 Report

This work was an interesting review of sRNAs in Pseudomonas and Burkholderia spp. While I felt that the manuscript could be condensed and still convey the same material, I think it is perfectly acceptable for publishing following some lite editing and modification of the inserted Fig. 1. 

A few thoughts:

1. The introductory paragraphs throughout the review are usually well written and provide the

necessary information for subsequent sections that are often more technical. For example, in

section 1 (Introduction: what are bacterial sRNAs?), I appreciated the level of detail the authors

used to describe general features of bacterial sRNAs. Additionally, they effectively described the

molecular mechanisms by which these sRNAs modulate gene expression. Also, it was useful that

the authors mentioned cases in which sRNAs act differently from general trends. This effective

writing trend carried on throughout most of the introductory sections in the review.

2. One issue I found in the article came in the 3 rd paragraph of section 3 (lines 187-199).

The authors describe a study that used an in-silico approach to identify sRNAs in two strains of

P. aeruginosa. This study suggested that even though many sRNAs are predicted to be in both

strains’ genomes, because not all sRNAs can be detected, those transcripts must be strain

specific. The authors don’t mention that those transcripts could be only transcribed under certain

environmental conditions. This seems to be a critical oversight because the two strains sampled

were recovered from different disease contexts (could disease context, and not strain per se, be

the reason they see different expression?) Additionally, in this paragraph, the authors don’t

provide any explanation as to the very different sRNA repertoire sizes between the two strains

reported in citation 41 and 42. Also in section 3, I would prefer the last 2 paragraphs omit the

significant amount of technical detail as it is provided in subsequent sections.

3. In the file shared for review, Figure 1 is cut off extensively. I imagine this was not intentional but due to some formatting issue.

4. What does "Type" indicate in Table 1? regulators and targets are already listed so it is a bit unclear. Perhaps add this to the table legend?

line 212: "underling" should be "underlying"

Author Response

REBUTTAL LETTER

Point-by-point answer to the questions raised by the reviewers

Reviewer #1

This work was an interesting review of sRNAs in Pseudomonas and Burkholderia spp. While I felt that the manuscript could be condensed and still convey the same material, I think it is perfectly acceptable for publishing following some lite editing and modification of the inserted Fig. 1. 

A few thoughts:

1. The introductory paragraphs throughout the review are usually well written and provide the necessary information for subsequent sections that are often more technical. For example, in section 1 (Introduction: what are bacterial sRNAs?), I appreciated the level of detail the authors used to describe general features of bacterial sRNAs. Additionally, they effectively described the molecular mechanisms by which these sRNAs modulate gene expression. Also, it was useful that the authors mentioned cases in which sRNAs act differently from general trends. This effective writing trend carried on throughout most of the introductory sections in the review.

ANSWER: Thank you very much for your positive appreciation. We sincerely appreciated your comments on our efforts to put together a document which we think is useful for the journal readers.

2. One issue I found in the article came in the 3 rd paragraph of section 3 (lines 187-199). The authors describe a study that used an in-silico approach to identify sRNAs in two strains of P. aeruginosa. This study suggested that even though many sRNAs are predicted to be in both strains’ genomes, because not all sRNAs can be detected, those transcripts must be strain specific. The authors don’t mention that those transcripts could be only transcribed under certain environmental conditions. This seems to be a critical oversight because the two strains sampled were recovered from different disease contexts (could disease context, and not strain per se, be the reason they see different expression?)

ANSWER: Thank you for the correction. We have changed the text to accommodate your comment. New lines 195-196 now reads as follows: “…expression could be strain-specific or environmental-dependent.”

Additionally, in this paragraph, the authors don’t provide any explanation as to the very different sRNA repertoire sizes between the two strains reported in citation 41 and 42.

ANSWER: Thanks for the comment. Some comments from the authors of these studies point out technical issues related to the techniques used as one of the explanations for the different repertoire sizes. This kind of explanation is, in our view, out of the context of the present review and therefore we decided not to include it.

Also in section 3, I would prefer the last 2 paragraphs omit the significant amount of technical detail as it is provided in subsequent sections.

ANSWER: We partially agree that some info contained in the 2 mentioned paragraphs are further detailed in following sections. However, the simple delete would have as a consequence the loss of general information which we consider relevant. Therefore, we have decided to keep the two sentences.

3. In the file shared for review, Figure 1 is cut off extensively. I imagine this was not intentional but due to some formatting issue.

ANSWER: We have gone to webpage of the journal and had accessed to the word file and to the pdf file. While the figure is OK in the word file, it is extensively cut off in the pdf version. We will look carefully at the proof reading stage to guarantee that the right figure is published.

4. What does "Type" indicate in Table 1? regulators and targets are already listed so it is a bit unclear. Perhaps add this to the table legend?

ANSWER: The word “Type” was used to distinguish the cis and trans-encoded sRNAs. It is not the best choice, and therefore we have substituted “Type” by “Category” and included in the table legend an explanation for the word category: “chromosome location related to the mRNA target”.

line 212: "underling" should be "underlying"

ANSWER: Corrected. Our mistake.

Reviewer 2 Report

It is an interesting and well written review that compiles data on sRNAs involved in the virulence of two major bacteria species colonizing the respiratory tract of cystic fibrosis patients.

Two minor points:

line 322: S. Typhimurium should be in italic

line 480: Addition instead addiction

Author Response

REBUTTAL LETTER

Point-by-point answer to the questions raised by the reviewers

Reviewer #2

It is an interesting and well written review that compiles data on sRNAs involved in the virulence of two major bacteria species colonizing the respiratory tract of cystic fibrosis patients.

Two minor points:

line 322: S. Typhimurium should be in italic

ANSWER: Typhimurium should not be italic, but S. should be italic. We maintained the original formatting.

line 480: Addition instead addiction

ANSWER: Done. Our mistake.